# Assessment of transmitral and left atrial appendage flow rate from cardiac 4D-CT

Sophia Bäck [1,2], Lilian Henriksson[2,3], Ann F. Bolger[1,4], Carl-Johan Carlhäll[1,2,5], Anders Persson [2,3], Matts Karlsson [2,6] & Tino Ebbers [1,2✉]

**Abstract**

**Background** Cardiac time-resolved CT (4D-CT) acquisitions provide high quality anatomical images of the heart. However, some cardiac diseases require assessment of blood flow in the heart. Diastolic dysfunction, for instance, is diagnosed by measuring the flow through the mitral valve (MV), while in atrial fibrillation, the flow through the left atrial appendage (LAA) indicates the risk for thrombus formation. Accurate validated techniques to extract this information from 4D-CT have been lacking, however.

**Methods** To measure the flow rate though the MV and the LAA from 4D-CT, we developed a motion tracking algorithm that performs a nonrigid deformation of the surface separating the blood pool from the myocardium. To improve the tracking of the LAA, this region was deformed separately from the left atrium and left ventricle. We compared the CT based flow with 4D flow and short axis MRI data from the same individual in 9 patients.

**Results** For the mitral valve flow, good agreement was found for the time span between the early and late diastolic peak flow (bias: <0.1 s). The ventricular stroke volume is similar compared to short-axis MRI (bias 3 ml). There are larger differences in the diastolic peak flow rates, with a larger bias for the early flow rate than the late flow rate. The peak LAA outflow rate measured with both modalities matches well (bias: −6 ml/s).

**Conclusions** Overall, the developed algorithm provides accurate tracking of dynamic cardiac geometries resulting in similar flow rates at the MV and LAA compared to 4D flow MRI.

**Plain language summary**

Assessing the blood flow inside the heart is important in diagnosis and treatment of various cardiovascular diseases, such as atrial fibrillation or heart failure. We developed a method to accurately track the motion of the heart walls over the course of a heartbeat in three-dimensional Computed Tomography (CT) images. Based on the motion, we calculated the amount of blood passing through the mitral valve and the left atrial appendage orifice, which are markers used in the diagnostic of heart failure and assessment of stroke risk in atrial fibrillation. The results agreed well with measurements from 4D flow MRI, an imaging technique that measures blood velocities. Our method could broaden the use of CT and make additional exams redundant. It can even be used to calculate the blood flow inside the heart.

[1] Unit of Cardiovascular Sciences, Department of Health, Medicine and Caring Sciences, Linköping University, Linköping, Sweden. [2] Center for Medical Image Science and Visualization (CMIV), Linköping University, Linköping, Sweden. [3] Department of Radiology, and Department of Health, Medicine and Caring Sciences, Linköping University, Linköping, Sweden. [4] Department of Medicine, University of California San Francisco, San Francisco, CA, USA. [5] Department of Clinical Physiology in Linköping, and Department of Health, Medicine and Caring Sciences, Linköping University, Linköping, Sweden. [6] Department of Management and Engineering, Linköping university, Linköping, Sweden. ✉email: tino.ebbers@liu.se

In recent years, assessment of the cardiac physiology with time resolved computed tomography (4D-CT) has become more common, due to improved techniques for acquisition of time resolved data and decreasing radiation doses with modern equipment. Using iodine-based contrast agents in 4D-CT, the blood pool in the heart is clearly distinguishable, thus 4D-CT data provide excellent anatomical assessment of the beating heart. The possibility to extract the end diastolic and end systolic ventricular volumes, stroke volume and ventricular ejection fraction has been shown in many studies and are well summarized by Kaniewska et al.[1]. These conventional parameters of cardiac function may not fully address the stage or prognosis of many cardiac diseases. The ratio between early and late diastolic mitral inflow velocities is central to the diagnosis and assessment of diastolic dysfunction.

Echocardiography Doppler enables estimation of left atrial appendage (LAA) flow velocities in order to assess the risk of clot formation, and embolic events especially in patients with atrial fibrillation (AF). However, this is typically done using transesophageal echocardiography (TEE), which is invasive and often requires sedation. The volume flow through the mitral valve (MV) and the left atrial appendage can, in theory, also be obtained from morphological imaging data by extracting the volume change of the main heart chambers in every frame[2]. These volume-based approaches require high resolution 3D data with good contrast between blood pool and surrounding to achieve accurate delineation of the blood pool and myocardium, valves, papillary muscles, and myocardial trabeculae, which can be provided by 4D CT. This approach does not provide information on velocities, but quantification of flow rates. However, it was found that flow-rate derived indices correlate better with age and left ventricular remodeling compared to peak velocity based indices[3].

Segmentation of the data is a cumbersome and highly user-dependent process, however. Using an algorithm to track the motion based on only one manual segmentation allows for more automatic processing of the data and would also facilitate calculations of other parameters relative to the intracardiac blood flow dynamics[4,5].

When tracking the endocardial motion of the left heart, each region presents its own challenges. In the left ventricle (LV) the lateral wall is markedly trabeculated, making the true endocardial surface difficult to segment and track reliably. In the left atrium (LA), the most challenging region is the LAA, which is not only trabeculated but also contracts very rapidly during late ventricular filling. Tracking this contraction in CT images is demanding because cardiac motion is commonly reconstructed to 10–20 frames per cardiac cycle. As a result, only one or two images will capture the minimal LAA volume. Motion tracking of the left heart must accommodate both anatomic and dynamic challenges.

There are different approaches when tracking motion from three-dimensional images. One possibility is to estimate a deformation field that covers all voxels of one image and that, when applied to the source image, minimizes contrast differences between the source image and the target image[4]. This approach was used by Al-Issa et al.[6] to track the motion of the left atrium and the LAA in AF patients with and without a prior history of stroke. They found that the maximum surface area increase of the LAA was lower in patients with prior history of stroke than in patients without previous strokes. Alternatively, instead of tracking the full image, the interface between blood and cardiac muscle can segmented and the registration be focused on the interface. This introduces an additional processing step, but the registration is less computationally demanding and can be more easily adopted for specific regions. Vedula et al. used this method by extracting points that lay on the blood-muscle interface and calculated the deformation field diffeomorphic registration

algorithm, known as large deformation diffeomorphic metric mapping (LDDMM)[7]. Similar approaches have been used by other, as for instance Otani et al.[8] and García-Villalba et al.[9], who used a coherent point drift algorithm. However, the algorithms developed in these studies were not validated with data from other modalities.

Besides 4D-CT, the flow in the beating heart can also be assessed using 4D flow MRI, which, in contrast, directly measures the blood flow. It has become an important clinical tool in the hemodynamic assessment of a wide range of cardiac diseases, from congenital heart diseases, valvular diseases, shunts to cardiomyopathies[10,11]. The technique has also been used to assess blood flow in left atria of atrial fibrillation patients, demonstrating lower velocities in patients compared to healthy controls[12]. However, a 4D flow MRI acquisition is an average of many heart beats, has limited spatial resolution and is less sensitive to low velocities. These limitations especially hamper the assessment of flow in the LAA, because it is relatively small and low velocities are of special interest since they promote thrombus formation. Although it is difficult to assess flow patterns inside the LAA with 4D flow MRI, peak flow jets at the LAA orifice are clearly distinguishable, which facilitates quantification of peak flow rates. The reproducibility of LAA flow assessment with 4D flow MRI has not been studied before, but the LAA is in a similar location and has a comparable diameter as the pulmonary veins, which can be reliably assessed using 4D flow MRI[13]. Flow rates through the mitral valve can be accurately quantified using 4D flow MRI[3].

In this paper we calculate the transmitral and LAA flow rate from 4D CT using a motion tracking algorithm. We compared the flow rates derived from 4D-CT with 4D flow MRI derived LAA and mitral flow rates and cine MRI derived LV stroke volume.

## Methods

**Study population.** In this study we included 14 patients that underwent a clinical coronary CT angiography examination for diagnosis of chest pain. The participants agreed to take part in an additional MRI examination. The study was approved by the regional ethical review board in Linköping and written informed consent was obtained by all patients. The MRI measurement was usually performed within 2 h after the CT acquisition. Inclusion criteria for the study were a successful CT angiography acquisition, no contradictions to MRI, no atrial fibrillation, and less than mild valvular regurgitation. Of the 14 patients initially included, one patient was excluded due to a large difference in heart rate during CT and MRI studies, and 4 patients were excluded because parts of the LAA were missing in the CT. Consequently, 9 patients were analyzed in this study.

Due to the clinical question, the patient group was quite diverse, with two patients having wall motion abnormalities. Patient 1 had mildly to moderately depressed global LV systolic function. Patient 9 had regional hypokinesia in a smaller area basal inferoseptal.

**CT and MRI acquisition.** The CT images were acquired on a third-generation Siemens dual-source CT scanner (Somatom Force, Siemens Healthineers, Forchheim, Germany) during sinus rhythm and end-inspiration breath hold. The images were reconstructed to 20 temporal phases, equivalent to 0 to 95% of the R-R interval. The in-plane resolution was $0.353 \times 0.353$ mm $\pm$ 0.035 and the slice thickness was 0.5 mm with a 0.25 mm increment. The mean effective dose, calculated based on the European Guidelines for multislice CT, was 6.38 mSv[14].

For each patient, 4D flow MRI images, short axis cine balanced steady-state free precession (bSSFP) images, and three- and four-

**Table 1 Scan parameters.**

| Parameter | | Value |
|---|---|---|
| CT | | |
| | Detector Collimation (mm) | 192 × 0.6 |
| | Gantry rotation time (s) | 0.25 |
| | Pitch | 0.17–0.28 |
| | Quality reference (mAs) | 276 |
| | Reference tube voltage (kV) | 100 |
| 4D Flow MRI | | |
| | Velocity encoding range (cm/s) | 120 |
| | Flip angle (degree) | 5 |
| | Echo time (ms) | 2.9 |
| | Repetition time (ms) | 5.0 |
| | k-space segmentation factor | 2 |

chamber long-axis images were additionally acquired on a clinical 3.0-T Philips Ingenia MRI unit (Phillips Healthcare, Best, the Netherlands). The 4D flow images were acquired during sinus rhythm and free breathing using navigator respiratory gating with an effective temporal resolution of 0.04 s. The acquired spatial resolution was $2.9 \times 2.9 \times 2.9$ mm$^3$ and the cardiac cycle was reconstructed to 40 phases.

Detailed information on the CT and 4D flow MRI scanner settings can be found in Table 1. The bSSFP images were recorded during end-expiratory breath holds, with a reconstructed in-plane resolution of $1 \times 1$ mm$^2$, a slice thickness of 8 mm and 30 time frames covering the cardiac cycle. The echo time was between 1.4 and 1.6 ms, repetition time was 2.7–3.2 ms and the flip angle was 45 degree.

**Motion tracking and flow calculation.** The motion of the endocardium was tracked from the 4D CT data using an algorithm based on the optimal step nonrigid iterative closed point algorithm developed by Amberg et al.[15]. This algorithm deforms a source surface onto a target surface through reducing the distance between the surfaces while keeping the topology of the source. We modified this algorithm so that the LAA was registered separately from the LA to be able to capture the full motion of the LAA, see Fig. 1a.

To generate the source surface, the end diastolic time frame was manually segmented using ITK-SNAP[16]. Both aortic and mitral valve were segmented as being open, and the leaflets were not segmented. The segmentation separated the larger papillary muscles from the ventricular blood pool. The segmented surface was then further smoothed using Ansys SpaceClaim (Ansys, Canonsburg, Pennsylvania, USA) and the surface triangles were regularized to a size of 1.7 mm at the LV and LA and smaller triangles in the region of the LAA. The LAA was marked in this surface to allow for separate registration.

To generate the target surfaces, the blood pool was segmented automatically in the CT images based on a threshold of 250 Hounsfield units[17] using in-house code developed in MevisLab (MeVis Medical Solutions AG, Bremen, Germany). Then, the surface of the endocardium in the left atrium and left ventricle was manually extracted and the pulmonary veins and the aorta were cut in the same position in all time frames. In each timestep, the LAA was manually extracted for separate registration. The manual preparation took ca 60–120 min per patient, which can be reduced considerably using further automatization and machine learning.

The registration was started at 0% RR (the source surfaces) and performed successively forward as well as backward in time. So, the geometry at 0% RR (the source surface) was registered to 5% RR and 95% RR, which were then registered to 10% RR and 90%

RR, respectively, and repeated until 50% RR was reached from both branches. The position of 50% RR was interpolated between forward and backward results (Fig. 1).

The algorithm was implemented in MATLAB R2019b (The MathWorks, Inc., Natick, Massachusetts, USA). Tracking the motion over the full cardiac cycle took ca 2–3 h per patient on a workstation using an Intel Xeon E5-2630 CPU. For each time step, the surface of the LA and LV was registered first. The edge points at the pulmonary veins and the aorta, which are at the boundary of the tracked surface, were set to be non-moving, which eases the definition of boundary conditions in later computational fluid dynamics (CFD) simulations. This does not affect the registration of the LAA and LV motion, which are evaluated here. After this, the left atrial appendage was registered, with the ostium motion prescribed from the left atrial registration. Similar to Amberg et al., prior to the registration, the geometry was scaled to a $[1,1]^3$ cube[15].

To find the optimal deformation X of the source points, the registration algorithm minimizes the cost function $\bar{E}(X)$, which considers four different terms, see Eq. (1). X is a $4n \times 3$ matrix, where $n$ is the number of source points and each $4 \times 3$ sub-matrix describes the affine transformation of the corresponding point.

$$\bar{E}(X) = \left\| \begin{bmatrix} \alpha M \otimes G \\ WD \\ \beta D_L \\ \lambda D_{Tar} \end{bmatrix} X - \begin{bmatrix} 0 \\ WU \\ \beta U_L \\ \lambda U_{Tar} \end{bmatrix} \right\|_F^2 \quad (1)$$

The top row of Eq. (1) considers the stiffness of the source triangulation. The stiffness weight $\alpha$ regulates the stiffness, with a high value of $\alpha$ corresponding to a stiff mesh. At the start of the registration, $\alpha$ it is relatively stiff and then lowered with the progress of the optimization. The connectivity of source points is described in the node-arc incidence matrix $M$, which has one row per edge and one column per source point. The Kronecker product of the node-arc incidence matrix M, which defines the connection of the source nodes, and a weighting matrix $G := diag(1, 1, 1, \gamma)$, is multiplied with the stiffness weight $\alpha$. In this study, $\gamma$ was set to 1. The stiffness weight $\alpha$ was lowered from 100 to 1 in 100 steps for the registration of LA and LV and it was lowered from 100 to 10 in 20 steps for the registration of the LAA. Stiffness weights of 100 ensure that initially the algorithm covers global deformations while final stiffness weights of 1, respective 10, were chosen to avoid unstable solutions.

The second term regulates the distance between the transformed source triangulation and the target points. For each point on the source surface, whose coordinates are saved in the Matrix $D$, its nearest neighbor in the target is $U$. $W$ is a matrix to weight the points, it was set to zero for the landmark points and 1 for all other points.

The third term describes the landmark points. In this study, the landmark terms were used to freeze all boundary points and to predefine the motion of the LAA orifice when only registering the LAA. Therefore, the landmark weight $\beta$ was set to 10 for all iterations, which ensures that the points do not move. $D_L$ are the rows from $D$ that correspond to the Landmark vertices and $U_L$ are the matching Landmark positions.

To capture the large deformation of the LAA, a term for bi-directional registration was used. With this term, for a subsample of target vertices, the nearest neighbor in the source was sought (Fig. 1b). The target vertices were chosen so that no other vertex lay within a radius of about 5% of the overall size of the LAA. $U_{Tar}$ contains the positions of these target points and $D_{Tar}$ contains the positions of the nearest neighbors in the source. These points were weighted with a factor $\lambda$ of 0.5.

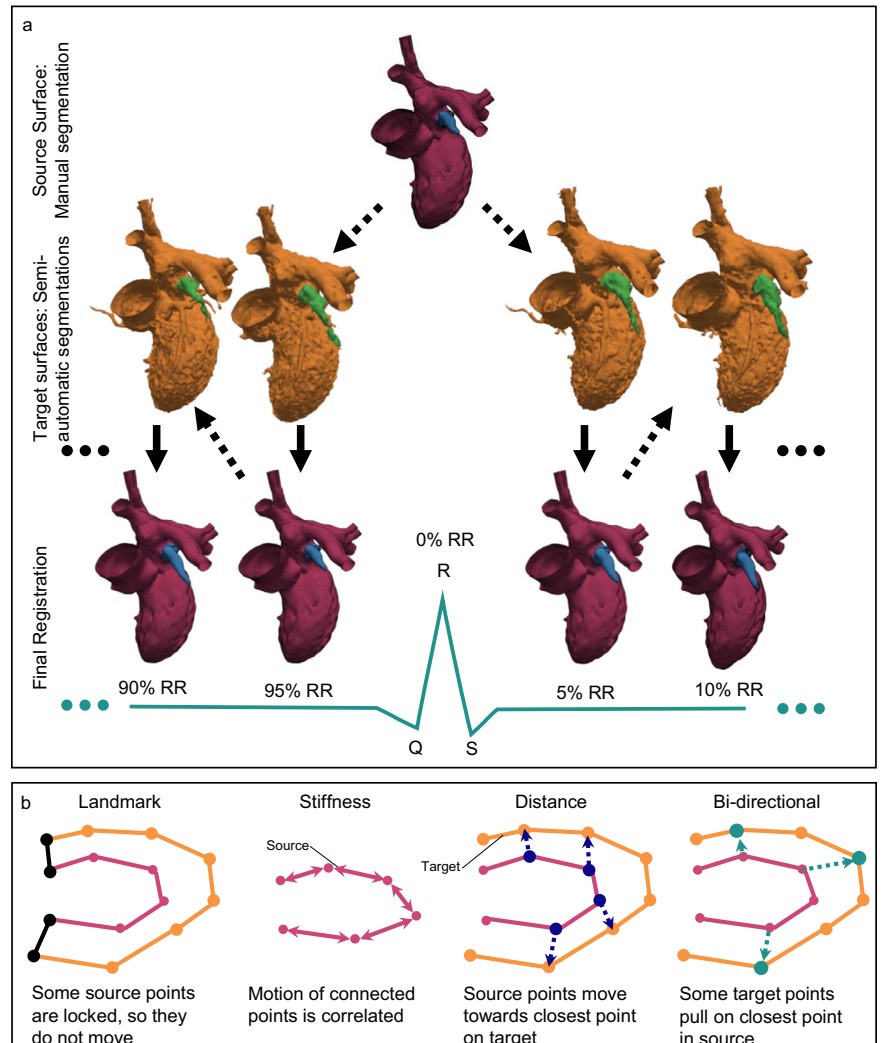

**Fig. 1 Registration Workflow. a** The Source surface at 0%RR was registered onto the target surfaces at 5%RR and 95%RR. The resulting final registrations were then registered onto 10%RR and 90%RR, respectively. The surface of the LAA (marked blue and green) was registered separately. **b** The registration was driven by four different terms: Landmark, Stiffness, Distance and Bi-directional.

To be able to use the derived motion in future CFD simulations, after the registration, the motion of each point was interpolated in time and space based on cubic B-splines with 15 knots, which was found to provide sufficient smoothing while preserving the characteristic of the data[18]. These splines were exported to a temporal resolution of 0.5 ms. Future CFD simulations would either only use the wall motion as a boundary condition, or, when simulating only the blood flow in the left atrium, the motion of the atrial wall combined with the flow rate through the mitral valve.

To calculate the mitral valve flow from the registration, the volume of the ventricle was calculated for every 5 ms using the divergence theorem and then differentiated with respect to time. Negative volume changes equate to the flow through the mitral valve. To obtain a LAA flow estimate from the CT based registration, a static plane was defined which separated the LAA from the LA over the complete cardiac cycle. This plane was then defined as the LAA orifice in each time step. This approach was chosen to match the flow measurements from 4D flow MRI.

**Comparison to 4D flow MRI**. The 4D CT based flow rates were compared to 4D flow MRI measurements of the same individual.

In the 4D flow MRI images, the mitral valve flow was calculated by valve tracking. The valve annulus movement was automatically tracked on a cine bSSFP (balance steady-state free-precession) image (3 chamber view) over the cardiac cycle, and from the resulting positions reformatted planes were placed in the 4D flow volume. Subsequently, the volumetric stroke volume was calculated by integrating the velocity over the mitral valve orifice, and correcting for through-plane motion, similarly to[19]. Measuring LAA flow from 4D flow MRI is challenging, as the LAA cannot be clearly defined. Therefore, the flow rate was measured in three parallel planes, that were set during LAA peak outflow phase and kept at the same position though the cardiac cycle. In every plane, the LAA orifice was defined by manual segmentations. The flow curves in the LAA orifice were calculated by averaging the volume flow from the three planes.

**Statistics and reproducibility**. The algorithm was applied on 9 subjects. Agreement of the two measurement modalities was measured with linear regression and with Bland-Altman analysis. Normally distribution was ensured using the Shapiro-Wilk test. All statistical tests were performed using Matlab 2019b.

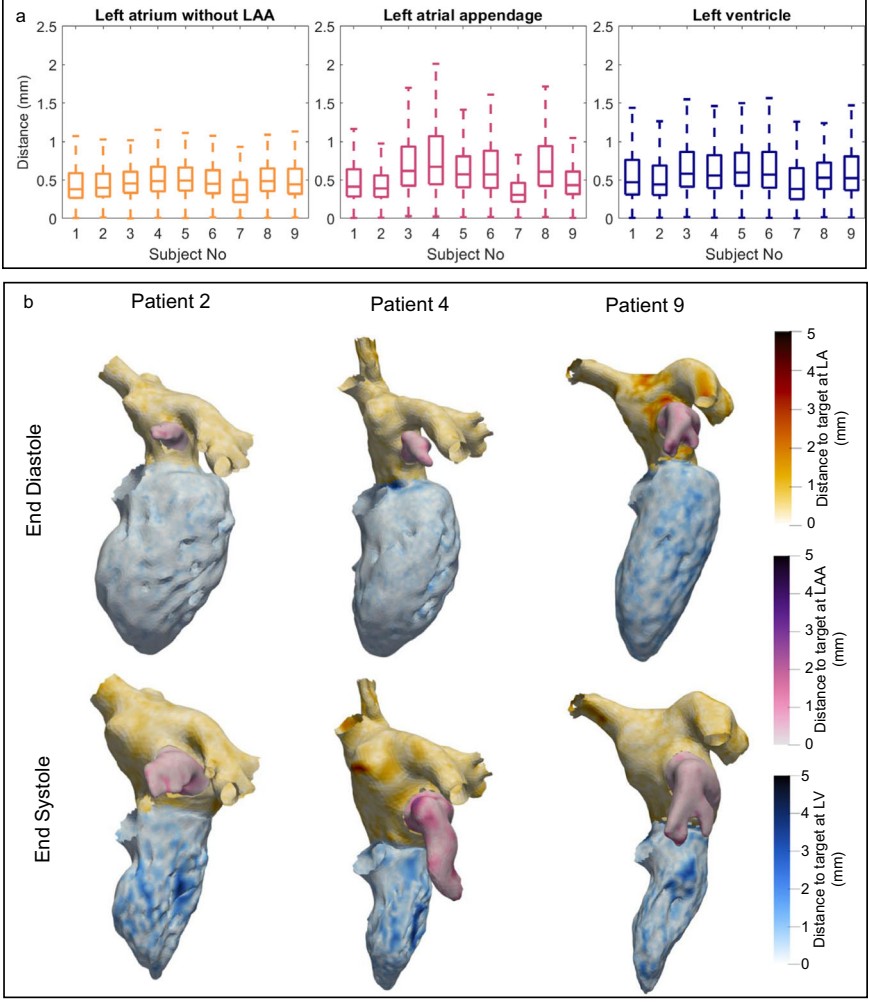

**Fig. 2 Evaluation of distance between registration and target surface. a** Boxplot diagrams of distance between registration and target surface. The central mark indicates the median, bottom and top edges of the box indicate the 25th and 75th percentiles, respectively. The whiskers extend to the most extreme data points not considered outliers, and the outliers are not shown for better visibility. **b** Visualizations of the distance from registration to target in 3 patients.

**Reporting summary**. Further information on research design is available in the Nature Portfolio Reporting Summary linked to this article.

## Results

**Registration evaluation**. To evaluate the quality of the motion tracking algorithm, the distance to the threshold-based target surfaces was calculated for each point in the registration, shown in Fig. 2a. Outliers are not displayed in the boxplot for better visibility. The spatial distribution of the distance between the registration and the target in three example patients is shown in Fig. 2b. Supplementary Movie 1 shows the distance to the target surface over time. In all regions, the wall motion was tracked closely with an average distance of 0.5 mm. The best results were obtained in the left atrium, followed by the LAA and the left ventricle.

**Mitral valve flow**. We compared the flow rates calculated from 4D-CT with 4D flow MRI measurements of the same individual. The flow through the mitral valve is characterized by two peaks, the early diastolic peak (E-wave) and the late diastolic peak (A-wave). For all 9 patients, a clear distinction of the two peaks is visible in both the 4D-CT based flow and the 4D flow MRI based flow (Fig. 3a). The time span between the early and late

diastolic peak flow is in good agreement (mean difference <0.01 s, standard deviation 0.1 s), see Fig. 3b. Furthermore, the area under the curves, which resembles the ventricular stroke volume, correlates well ($R^2 = 0.68$), with a bias of 14 ml. The stroke volume measured from CT was further compared to the stroke volume measured from short axis MRI, see Fig. 4. The correlation of this comparison was weaker ($R^2 = 0.46$), but the bias was only 3 ml, with all patients except for two being in the range of −2 to 15 ml. In one patient, the stroke volume measured with CT was 30 ml larger than with short axis MRI and in one patient it was 39 ml smaller.

For all patients except patient 6, the early diastolic peak flow was higher in the 4D-CT than in 4D flow MRI. The late diastolic peak flow correlates better (mean difference = 18 ml/s) than the early diastolic peak flow (mean difference = 165 ml/s). The Ef/Af ratio, which is the ratio of the early and late peak flow rates, was higher in the 4D CT data compared to 4D flow MRI.

**Left atrial appendage flow**. The flow rate though the LAA orifice is shown in Fig. 5a. In general, measuring the flow though the LAA orifice in 4D flow MRI is challenging, due to the lack of contrast in the LAA. However, the flow at peak contraction can be measured due to the relatively high velocities. The peak outflow rates match well between CT and 4D

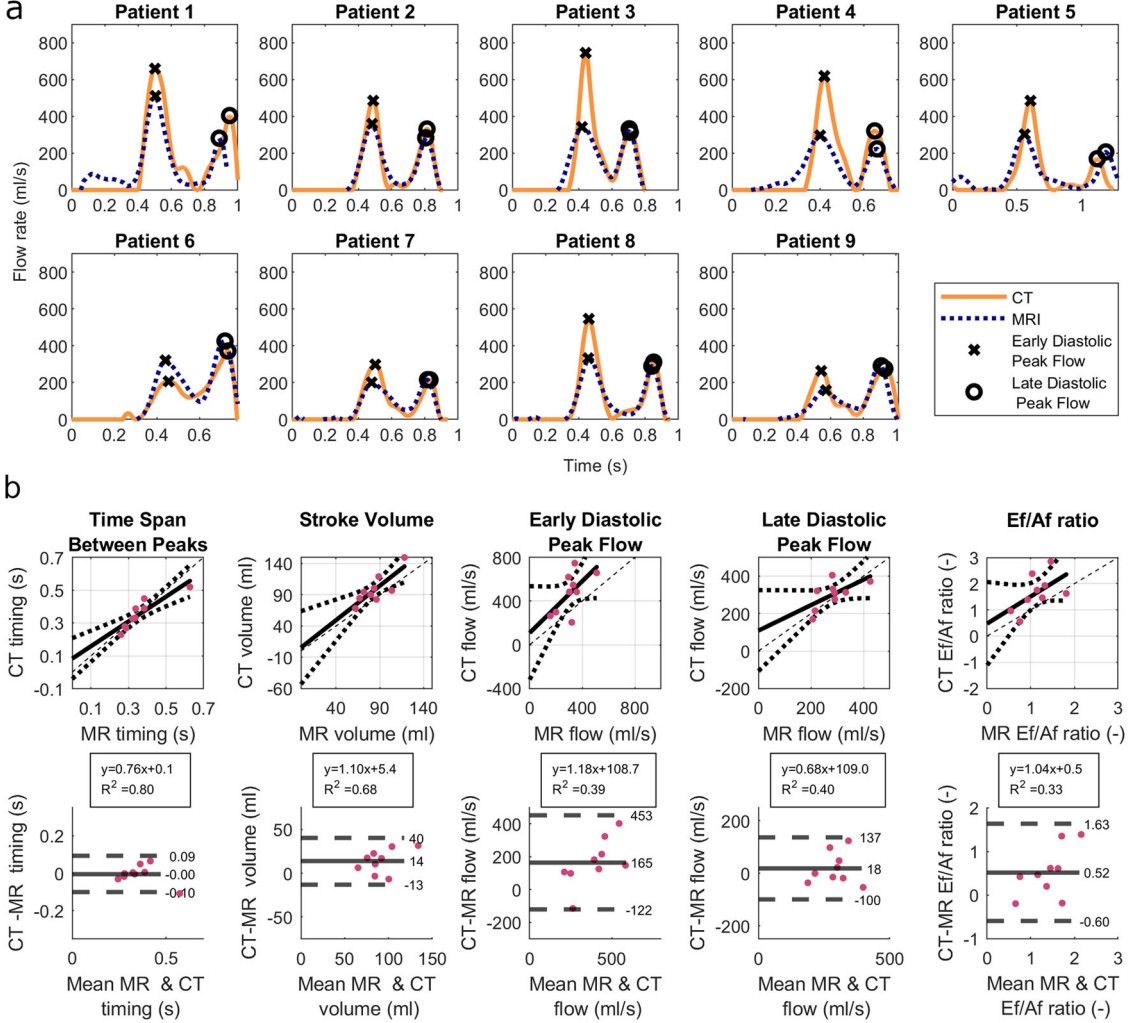

**Fig. 3 Mitral Valve Flow. a** Flow thought Mitral Valve calculated based on 4D flow MRI and time resolved CT, Early Diastolic Peak Flow is marked with x, and Late Diastolic Peak Flow is marked with o. **b** Comparison of MV flow parameters: Time Span Between Peaks, Ventricular Stroke Volume, Early Diastolic Peak Flow, Late Diastolic Peak Flow, and Ef/Af ratio in 4D flow MRI and 4D CT. Top: Solid line indicates Linear Regression line; Dotted lines indicate 95% confidence bounds; Dashed line indicates identity line. The box shows the linear regression equation and the coefficient of determination $R^2$. Bottom: Bland-Altman Plot; Solid line indicates the bias, Dashed lines indicate ±1.96 standard deviation.

flow MRI ($R^2 = 0.75$; Fig. 5b). The Bland-Altman analysis indicates a slightly higher flow rate measured by MRI than by CT (mean difference = −6 ml/s). In the CT based flow curve for patients 1–5 and Patient 8, the LAA flow not only shows a peak at late diastole, but also a peak at early diastole that corresponds to the early diastolic mitral valve flow. This finding is also clearly visible in the MRI data for patients 1 and 5 but is less pronounced for the other patients.

## Discussion

To measure the flow through the mitral valve and the left atrial appendage orifice from 4D CT, we developed and evaluated a surface-based algorithm. The algorithm successfully tracked the endocardial motion and the processing pipeline allowed for local adjustment of the approach, which was utilized to optimize the tracking of the LAA. The LAA flow at peak contraction and the ventricular stroke volume were in good agreement with flow measurement from 4D flow MRI and LV stroke volume from cine MRI.

The tracking algorithm obtained highest agreement between the registered and the target surface for the smooth atrial wall, but

agreement was still good for the more trabeculated wall of the LAA and LV.

The flow through the mitral valve showed good agreement for the time interval between the early and late diastolic peak flow, as well as for the stroke volume for CT and 4D flow MRI. This is promising, considering that the CT and 4D flow MRI acquisition are two separate examinations that were performed within a few hours. We found a good correlation between 4D-CT and 4D flow MRI measurements of the ventricular stroke volume with a mean difference between the two measurements of 14 ml. Furthermore, we compared the stroke volume measured with CT to the stroke volume measured in short axis MRI images, which is the preferred clinical method for LV volume assessment[20] and found a bias of 3 ml, with 7 of 9 patients ranging from -2 to 15 ml. In one patient, the stroke volume measured with CT was 30 ml larger than with short axis MRI and in one patient it was 39 ml smaller. This bias is similar to a prior meta-analysis comparing ventricular stroke volume measured with cardiac CT and MRI which found a mean difference of 3 ml as well[1]. The stroke volume measured with 4D flow MRI was 11 ml less compared to short axis MRI using a volume-based approach. Kamphuis et al. also compared stroke volume measurements using 4D flow MRI and short axis

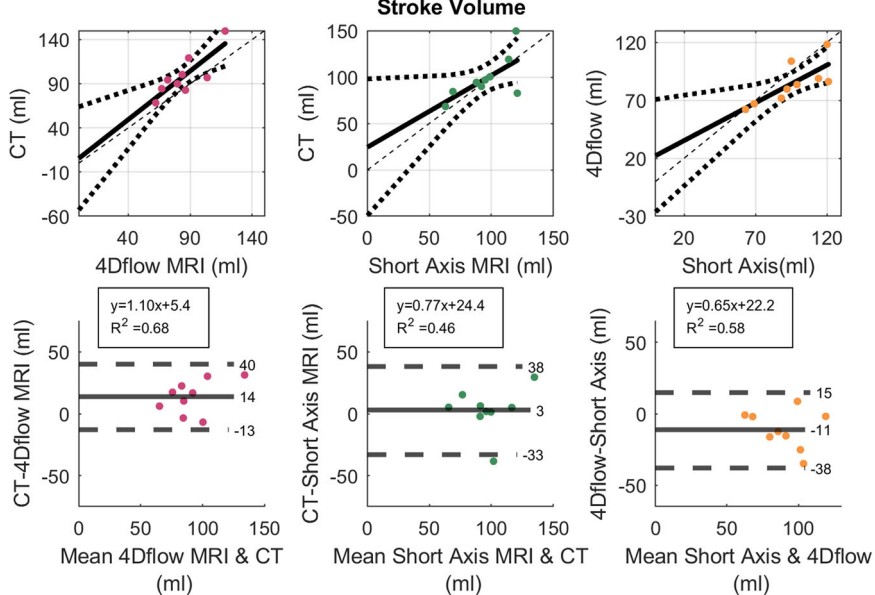

**Fig. 4 Comparison of Ventricular Stroke Volume measured with CT, 4D flow MRI and Short Axis MRI.** In the comparison of CT and Short Axis MRI, the differences between the measurements were not normal distributed, so the limits of agreement might be overestimated Top: Solid line indicates Linear Regression line; Dotted lines indicate 95% confidence bounds; Dashed line indicates identity line. The box shows the linear regression equation and the coefficient of determination $R^2$. Bottom: Bland-Altman Plot; Solid line indicates the bias, Dashed lines indicate ±1.96 standard deviation.

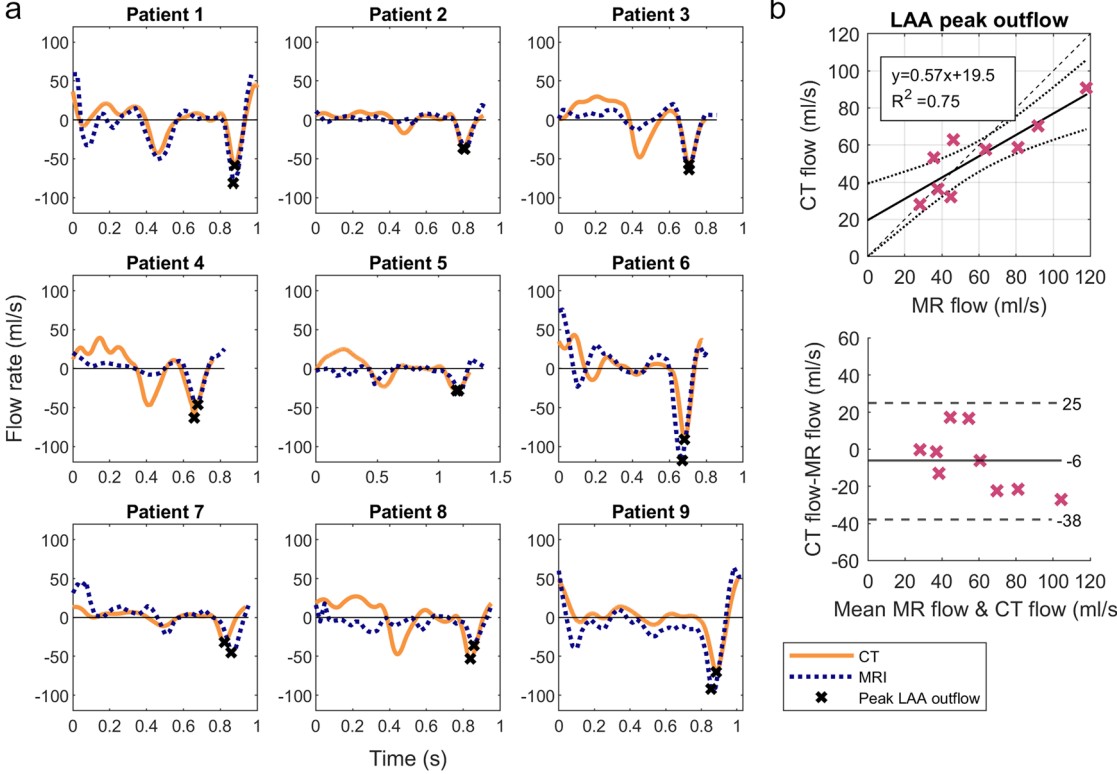

**Fig. 5 LAA Flow Rate. a** Flow rate though LAA orifice calculated based on 4D flow MRI and time resolved CT, peak LAA outflow is marked with X. **b** Comparison of Peak LAA Outflow Rate in 4D flow MRI and 4D CT. Top: Linear Regression: Solid line indicates Linear Regression line; Dotted lines indicate 95% confidence bounds; Dashed line indicates identity line. The box shows the linear regression equation and the coefficient of determination $R^2$. Bottom Bland-Altman Plot, Solid line indicates the bias, Dashed lines indicate ±1.96 standard deviation.

MRI and found the stroke volume to be 7 ml less in 4D flow MRI[20]. This indicated that the LV stroke volumes computed from 4D CT in this study are comparable to what has been found in earlier studies.

The LAA peak outflow from CT correlated well with 4D flow MRI findings. Some patients also showed an early diastolic peak in the flow rate, which was more prominent in the CT data than in the MRI data. The early diastolic peak in the LAA flow has also been shown using transesophageal echocardiography[21]. The flow rates through the LAA orifice were computed using a static plane in CT in order to facilitate the comparison with 4D flow MRI, and thus both measurements include the LAA contraction and overall motion. As the algorithm computes the wall motion from CT, it is also possible to use a moving plane to compute flow rates in the LAA. In future studies that do not include a comparison to 4D flow MRI, but rather relate the LAA flow rates to other clinical parameters, this is probably the more desired approach. The flow rates during late diastole agree better than during early diastole, both in the LAA and in the mitral valve. For patient 3, 4, and 8, the CT based method shows a peak in the LAA contraction during early diastole, which is not visible in 4D flow MRI. For these patients, the E peak of the MV flow curve is also higher in the CT based method than in 4D flow MRI. Alattar et al. also found lower MV flow rates in 4D flow MRI compared to 2D flow MRI, however, they found similar agreement between early and late diastolic flow[3]. One possible explanation for the difference between early and late diastole could be that 4D MRI is computed as the average of many heart beats, that are composed to one image based on the R wave in the ECG signal, which corresponds to end diastole. Each measured cardiac cycle is normalized to the average heart rate to compose the image. At low heart rates, changes in the heart rate mainly alter the length of diastasis, which then alters the relative position of the E wave in the heart rate-normalized signal, which could lead to a stronger smoothing of the signal during this time.

Another possible explanation is that the 4D flow MRI was acquired during free breathing, while the CT was acquired during an inspiration breath hold and the short axis MRI was acquired during end-expiration breath hold. This could alter the pressure in the lungs, which influences the pressure in the left atrium and might alter the flow through the mitral valve.

Due to the high spatial resolution of the CT image and the good contrast between blood pool and myocardium, the segmentations do not consider the papillary muscles to be part of the blood pool. In contrast, the short-axis MRI volume-based approach considers the papillary muscles to be part of the blood pool, leading to larger end diastolic and end systolic volumes compared to CT. Both methods computed a higher stroke volume than 4D flow MRI, and in general, this results in higher flow rates. It is possible that some of the trabeculated structures appeared as blood pool during end diastole and myocardial during the end systolic phase resulting in a slightly increased estimated stroke volume from CT and short axis MRI.

Due to the larger peak E flow computed from CT compared to 4D flow MRI, the Ef/Af ratio was higher in the CT based computation compared to 4D flow MRI, with a bias of 0.52. This bias is relatively large considering the range of Ef/Af ratios measured. For future studies, it would be important to evaluate the accuracy of the Ef/Af ratio more in detail.

Previous studies have assessed the surface area and function of the LAA based on time-resolved CT, but without comparison to flow or other measurements obtained with a different imaging modality in the same subject. Otani et al. tracked the atrial motion in three patients, but in one of the three patients the algorithm did not match the minimum LAA volume. The authors speculate that this was due to a portion of the LAA being excluded from the CT acquisition. Our CT data as well as MRI flow data suggest, however, that in many patients the LAA contracts more rapidly than reported in these studies, and that the small LAA volume at 0% RR could have been a result of this contraction. As shown in Supplementary Fig. 1, following our registration approach the surface area of the LAA increases by a factor of 2–3.5 over the cardiac cycle in all 9 patients. Otani et al. measured a maximum 1.2-to-1.5-fold increase among their 3 subjects. Al-Issa et al.[6] investigated LAA contraction in 36 atrial fibrillation patients referred for catheter ablation. They found an area increase factor of $1.24 \pm 0.33$ in patients with a prior history of stroke and an area increase factor of $1.53 \pm 0.41$ in patients without a prior history of stroke. The area change calculated in this study is larger in all subjects than reported in previous studies. This could result from the patient cohort, since Al-Issa et al. investigated patients with atrial fibrillation, or the analysis approach. Our comparison with 4D flow MRI data in the same individual, which measured similar or even higher peak LAA outflow rates, suggests that our tracking algorithm accurately depicts the actual LAA motion. Previous studies have not included a validation of the analysis technique.

The main limitation of this study is the rather small number of patients. Including a larger number of patients is challenging, due to the relatively time-consuming analysis, the long duration of the 4D flow examination, and logistical complexities on examining patients with both 4D CT and 4D flow MRI within a short period of time. The wall motion algorithm required manual adjustment of the semi-automatic target surfaces, which is time-consuming and error prone. In the future, the processing could be automated and accelerated using machine learning. To track the relatively fast motion of the LAA, it was tracked separately, since then it can be deformed independently of the LA. This increased the processing time but was necessary to properly track the motion.

Unfortunately, there is no absolute gold standard when measuring flow rates in the heart. Velocities are clinically measured by ultrasound, but this does not provide information on flow rates. We have compared the flow rates derived from CT to flow rates measured with 4D flow MRI, and for the ventricular stroke volume, we also compared to stroke volume measurements from short axis MRI. By this we could identify similarities and differences, but larger clinical studies are necessary to evaluate if flow rates from CT or 4D flow MRI are more fitted to diagnose clinical conditions.

The method proposed here measures the volume change of the left ventricle and associates positive volume change with flow through the mitral valve. Therefore, it is not applicable to patients with aortic regurgitation since it would assign backflow through the aortic valve as flow through the mitral valve. Despite that, the method computes flow rates rather than velocities, which was found to be stronger correlated to LV remodeling than velocity-based markers[3].

When examining the LAA, this method has the advantage that it is non-invasive, in contrast to a transesophageal echocardiogram (TEE), which is currently recommended for assessment of the LAA. Extracting transmitral and LAA flow rate from CT could, in the future, reduce the need for using additional imaging exams in some patient groups, as, for instance, the TEE exam in patients with atrial fibrillation before ablation treatment. For patients with an indication for a CT angiography, as studied in this project, the addition of the presented approach would allow for accurate assessment of both diastolic and systolic ventricular function, besides the assessment of the coronary arteries, potentially removing the need to perform some ultrasound investigations. The derived wall motion can also be used as boundary conditions in CFD simulations of the heart, while the transmitral or LAA flow rates could facilitate simulation of the blood flow in

only the left atrium or LAA. García-Villalba et al.[9] recently compared CFD simulations in the left atrium using fixed-walls and moving walls. Results indicated that moving-wall simulations better risk-stratified their small cohort for LAA thrombosis, which stresses the importance of accurate wall motion estimation.

In conclusion, calculating the flow through the mitral valve and the LAA orifice from 4D-CT is feasible. This is supported by the good correlation of peak LAA outflow, ventricular stroke volume and time interval between the early and late diastolic peak flow compared to 4D flow MRI.

## Data availability
The data that support the findings of this study are available on reasonable request from the corresponding author to researchers who meet the criteria for access to confidential data. The source data is provided in Supplementary Data 1.

## Code availability
The custom computer code used to generate results that are reported in the paper are available under https://doi.org/10.17605/OSF.IO/BZ89C[22].

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

## Acknowledgements
This work was funded by the Swedish Medical Research Council grant number 2018-02779, the Swedish Heart and Lung Foundation grant numbers 20200220 and 20210441, ALF Grants, Region Östergötland grant number RÖ−940960, the Swedish Research Council grant number 2018-04454, Sweden's Innovation Agency Vinnova, project 2019-02261 and the County Council of Östergötland, grant number LIO-899441. The computations were enabled by resources provided by the Swedish National Infrastructure for Computing (SNIC), partially funded by the Swedish Research Council through grant agreement no. 2018-05973.

## Author contributions
S.B.: development of the algorithm; conception and design of the study, analysis and interpretation of data, drafting the article. L.H.: data acquisition, A.B.: conception and design of the study, C.C.: conception and design of the study, A.P.: conception and design of the study and data acquisition, M.K.: conception and design of the study, T.E.: conception and design of the study, analysis and interpretation of data, supervision.

## Funding

## Competing interests
All authors reviewed the manuscript critically and approved the final version. The authors declare no competing interests.
