## [Peer Review File · Communications Medicine]

Web links to the author's journal account have been redacted from the decision letters as indicated to maintain confidentiality.

17th Mar 22

Dear Professor Ebbers,

I sincerely apologise for the delay in getting back to you with a decision. Your manuscript entitled "Assessment of Mitral Valve and Left Atrial Appendage Flow Rate from Cardiac 4D-CT" has now been seen by 3 referees, whose comments are appended below. You will see from their comments copied below that while they find your work of potential interest, they have raised quite substantial concerns that must be addressed. In light of these comments, we cannot accept the manuscript for publication, but would be interested in considering a revised version that addresses these serious concerns.

It is our Editorial view that all Reviewers' concerns would need to be addressed in full. We hope you will find the referees' comments useful as you decide how to proceed. Should further experimental data or analysis allow you to address these criticisms, we would be happy to look at a substantially revised manuscript. However, please bear in mind that we will be reluctant to approach the referees again in the absence of major revisions. If the revision process takes significantly longer than six months, we will be happy to reconsider your paper at a later date, as long as nothing similar has been accepted for publication at Communications Medicine or published elsewhere in the meantime.

We are committed to providing a fair and constructive peer-review process. Do not hesitate to contact us if you wish to discuss the revision or if there are specific requests from the reviewers that you believe are technically impossible or unlikely to yield a meaningful outcome.

If you decide to submit a revised version, we ask that you ensure your manuscript complies with our editorial policies. Please see [our revision checklist](https://www.nature.com/documents/commsj-file-checklist.pdf) for guidance on formatting the manuscript and complying with our policies.

Communications Medicine seeks to improve the standards and transparency of reporting in our papers, and to ensure that all submissions conform with the editorial policies of Nature Research. When uploading your revised files please complete and submit [Reporting Summary](https://www.nature.com/documents/nr-reporting-summary.pdf) and [Editorial Policy](https://www.nature.com/documents/nr-editorial-policy-checklist.pdf) checklists as 'checklist' file types. Please note that these forms are a dynamic 'smart pdf' and must therefore be downloaded and completed in Adobe Reader, instead of being opened in a web browser. All points on the checklists must be addressed; if needed, please revise your manuscript in response to these points.

Your revised paper will not be returned to the editors for evaluation until these forms are provided.

Please use the following link to submit your revised manuscript, point-by-point response to the referees' comments (which should be in a separate document to any cover letter), reporting summary, editorial policy checklist and any additional files:
[link redacted]

We hope to receive your revised paper within six months. Please get in touch if you think you might need more time.

Please do not hesitate to contact us if you have any questions or would like to discuss the required

revisions further. Thank you for the opportunity to review your work.

Best regards,

Andreia Cunha, PhD
Chief Editor
Communications Medicine

on behalf of

Betty Raman, PhD
Editorial Board Member
Communications Medicine
orcid.org/0000-0002-1239-9608

Referee expertise:

Referee #1: 4D flow CMR and clinical applications

Referee #2: 4D flow CMR and computational modelling

Referee #3: Computational Fluid dynamics and 4D flow MRI

Reviewers' comments:

Reviewer #1 (Remarks to the Author):

Very interesting and robust study investigating the accuracy of transmitral flow hemodynamic curve generated by 4D-CT against 4D-Flow MRI. The study is well designed and reads nicely. I have a couple of questions that can improve the already well-written manuscript.

1) In the introduction I would suggest mentioning some of the limitations of purely volumetric-based indices we gain either from CT or CMR.

2) please provide the citation regarding the accuracy of LAA appendage flow with 4D-Flow MRI

3) Figure 3 nicely represents the evaluated method. Could the authors comment more on why the distance between the registration and the target was most successful in LA vs LAA vs LV?

4) For segmentation purposes, how were the intracardiac structures handled? MV, papillary muscles etc.

5) Were there any wall motion abnormalities that might have caused the registration distance to be higher in LV (or other structures?). Any non- supraventricular arrhythmias – LBBB / RBBB?

6) For figure 2, an important LV diastolic parameter is E/A ratio. Could the authors please provide Bland Altman graph / analysis for this metric as well?

7) CT vs MRI comparison shows reasonable agreement but the absolute “clinical” agreement seems to be considerably off particularly when it comes to peak E values. However, LAA peak flow data seems to show a better agreement. Could the authors speculate on why?

8) I would recommend to report all volumetric flow rate numbers in mL/s with no decimal points. 4D-Flow MRI which is a gold standard in this study does not have a spatial resolution good enough to be precise within these significant digits. mL are in my opinion sufficient.

9) Did the authors applied mitral valve ring motion tracking correction for the 4D-Flow MRI?

10) Can authors comment on internal reproducibility of the LAA flow rate from the 4D-Flow MRI datasets?

11) Transmitral flow evaluation is indeed an important marker used for many clinical stratifications. What is clinically more desirable using this technique, which is for image acquisition certainly faster than 4D-Flow, better than standard echo or MRI? What was the overall post-processing time for the 4D-CT?

12) Any chance these patients had Doppler or tissue Doppler datasets?

Reviewer #2 (Remarks to the Author):

This work employs image processing techniques to extract the flow rates through the mitral orifice (MO) and across left atrial appendage (LAA) from dynamic CT (3D+time) data. A comparison of these flow rates from CT data with 4D MRI-based measurements showed reasonable agreement in a small cohort of patients (n=9) although substantial differences could be found in the MO flow during late diastole.

Although the image segmentation and registration methods employed here are well established, the study demonstrates their utility in computing flow rates from dynamic CT data which, are otherwise, not available. The manuscript is well written and methods were adequately discussed. However, the limitations of the approach are not adequately discussed.

1. My understanding is that most patients undergoing CT also undergo ultrasound (US) evaluation. Therefore, the MO flow rate (and potentially flow across LAA?) could be easily derived from US measurements that offer a high temporal resolution. In that case, the motivation to compute these quantities from dynamic CT data should be strongly justified.

2. Both CT and MRI data employed in the current study are known to suffer from poor time resolution which could affect the computed time derivatives. Have the authors compared the computed flow rates against US data?

3. Although the study postulates that these CT-based flow measurements could be used as boundary conditions for computational fluid dynamics (CFD) models, a general CFD simulation set up for moving domain blood flow problems compute these flow rates as a by-product as opposed to employing them as boundary conditions. Applying these quantities as boundary conditions to a moving-domain problem may over constrain the problem in satisfying mass conservation.

4. The study computes mitral flow rate from the time rate of change of the ventricular volume during diastole. However, this limits the analysis to cases when the aortic valve is fully closed and will not apply to a pathological aortic valve such as when there is aortic valve regurgitation.

5. The present approach involves tedious segmentations and point-set registration techniques that can be prohibitively time-consuming. Can the authors discuss processing times for each dataset and the

translational applicability of this analysis?

6. Page 10, Discussion, while explaining the differences in stroke volume between CT and MRI compared to an earlier meta analysis, it is noted that the present MRI-based stroke volume is based on integrating flow rate through the mitral orifice and could potentially explain the large difference from CT data. This only raises concerns on the accuracy of the MRI-based velocities and flow rates as both methods should theoretically converge to the same value. On the other hand, can the authors use ventricular volumes from MRI data and compare the stroke volume against CT? This should be straightforward as the volumes could be easily extracted from 4D MRI. This poses two issues - ideally both methods should give exact results (volume-based or integral of flow rate). If the MRI-based volumes

7. The motivation to perform a separate registration of LAA is not discussed.

Minor:

1. Caption of Fig. 2 should be "Flow through..."
2. Figure captions need to be properly formatted for subfigures (A, B, etc.).
3. Several citations in the manuscript are not properly referenced. Please fix them.
4. Typo in Line 156, Page 10: "intervall"

Reviewer #3 (Remarks to the Author):

Please see the uploaded annotated manuscript with reviewer comments in red.

This manuscript details a method to calculate volumetric blood flow rates in the LAA and through the aortic valve by measuring changes in the volumes using 4D-CT data. They validate their method using 4D-Flow MRI.

4D-Flow MRI is itself not accurate since there is the issue of collecting data over multiple cardiac cycles and volume averaging in addition to issues of noise and image artifacts. The authors claim that the method is automated. However, when reading carefully through the manuscript it is evident that there is significant amount of manual work involved in segmenting the lumen surface. This is susceptible to operator variability.

There are also places in the manuscripts where parameters for interpolation, segmentation etc. are specified arbitrarily without any justification. For example, on line 299, I found the following "the motion of each point was interpolated in time and space based on cubic B-splines with 15 knots". There is no justification why the authors are using 15 knots in the B-spline.

Overall, I feel the authors should first have done controlled in vitro or even in silico experiments wherein the ground truth is known to validate the method. In this paper, they seem to compare their method to another that is error prone and then commented on their method's accuracy.

COMMSMED-21-0591-T

Assessment of mitral valve and left atrial appendage flow rate from cardiac 4D-CT

Response to Referees

Reviewer #1

Very interesting and robust study investigating the accuracy of transmitral flow hemodynamic curve generated by 4D-CT against 4D-Flow MRI. The study is well designed and reads nicely. I have a couple of questions that can improve the already well-written manuscript.

1) In the introduction I would suggest mentioning some of the limitations of purely volumetric-based indices we gain either from CT or CMR.

[Response]

We added this paragraph to the introduction (page 3 line 55-63):

The volume flow through the mitral valve (MV) and the left atrial appendage can, in theory, also be obtained from morphological imaging data by extracting the volume change of the main heart chambers in every frame. These volume-based approaches require high resolution 3D data with good contrast between blood pool and surrounding to achieve accurate delineation of the blood pool and myocardium, valves, papillary muscles, and myocardial trabeculae, which can be provided by 4D CT. This approach does not provide information on velocities, but quantification of flow rates. However, it was found that flow-rate derived indices correlate better with age and left ventricular remodeling compared to peak velocity based indices⁵.

Furthermore, we also extended the section on limitations in the discussion.

2) please provide the citation regarding the accuracy of LAA appendage flow with 4D-Flow MRI

[Response]

We have added a citation and explained it in the response to comment 10.

3) Figure 3 nicely represents the evaluated method. Could the authors comment more on why the distance between the registration and the target was most successful in LA vs LAA vs LV?

[Response]

We added this section to the second paragraph of the discussion (page 12 line 181-183):

The tracking algorithm obtained highest agreement between the registered and the target surface for the smooth atrial wall, but agreement was still good for the more trabeculated wall of the LAA and LV.

4) For segmentation purposes, how were the intracardiac structures handled? MV, papillary muscles etc.

[Response]

Smaller intracardiac structures were ignored in the segmentation. The mitral valve was considered open, and the leaflets were not segmented specifically. Large papillary muscles were not considered to be part of the blood pool. We included this information in the methods section (page 21 line 341-343):

Both aortic and mitral valve were segmented as being open, and the leaflets were not segmented. The segmentation separated the larger papillary muscles from the ventricular blood pool.

5) Were there any wall motion abnormalities that might have caused the registration distance to be higher in LV (or other structures?). Any non-supraventricular arrhythmias – LBBB / RBBB?

[Response]

There were no non-supraventricular arrhythmias or LBBB/RBBB in the patient cohort. Patient 1 had mildly to moderately depressed global LV systolic function, but no regionality. Patient 9 had regional hypokinesia in a smaller area basal inferoseptal. The registration distance values are similar for these patients compared to the other patients. We added this information to the methods section 'Study population' (page 18 line 304-306).

6) For figure 2, an important LV diastolic parameter is E/A ratio. Could the authors please provide Bland Altman graph / analysis for this metric as well?

[Response]

We added the Bland Altman and Linear regression analysis for the Ef/Af ratio, which is the ratio of the early and late mitral valve flow rates. Furthermore, we added a section on the Ef/Af ratio to the discussion (page 15 line 233-236).

7) *CT vs MRI comparison shows reasonable agreement but the absolute “clinical” agreement seems to be considerably off particularly when it comes to peak E values. However, LAA peak flow data seems to show a better agreement. Could the authors speculate on why?*

[Response]

We added this to the discussion (Page 13/14 line 203 to line 224):

The flow rates during late diastole agree better than during early diastole, both in the LAA and in the mitral valve. For patient 3, 4 and 8, the CT based method shows a peak in the LAA contraction during early diastole, which is not visible in 4D flow MRI. For these patients, the E peak of the MV flow curve is also higher in the CT based method than in 4D flow MRI. Alattar et al. also found lower MV flow rates in 4D flow MRI compared to 2D flow MRI, however, they found similar agreement between early and late diastolic flow⁵. One possible explanation for the difference between early and late diastole could be that 4D MRI is computed as the average of many heart beats, that are composed to one image based on the R wave in the ECG signal, which corresponds to end diastole. Each measured cardiac cycle is normalized to the average heart rate to compose the image. Changes in the heart rate alter the length of diastasis, which then alters the relative position of the E wave in the heart rate-normalized signal, which could lead to a stronger smoothing of the signal during this time.

Another possible explanation is that the 4D flow MRI was acquired during free breathing, while the CT was acquired during an inspiration breath hold and the short axis MRI was acquired during end-expiration breath hold. This could alter the pressure in the lungs, which influences the pressure in the left atrium and might alter the flow through the mitral valve.

8) *I would recommend to report all volumetric flow rate numbers in mL/s with no decimal points. 4D-Flow MRI which is a gold standard in this study does not have a spatial resolution good enough to be precise within these significant digits. mL are in my opinion sufficient.*

[Response]

This is a good advice. We changed this in the figures and text.

9) Did the authors applied mitral valve ring motion tracking correction for the 4D-Flow MRI?

[Response]

Yes, the motion of the mitral valve ring was tracked for the 4D flow MRI. We clarified this process in the methods section (page 25 line 416-421):

The valve annulus movement was automatically tracked on a cine bSSFP (balance steady-state free-precession) image (3 chamber view) over the cardiac cycle, and from the resulting positions reformatted planes were placed in the 4D flow volume. Subsequently, the volumetric stroke volume was calculated by integrating the velocity over the mitral valve orifice, and correcting for through-plane motion, similarly to ¹⁹.

10) Can authors comment on internal reproducibility of the LAA flow rate from the 4D-Flow MRI datasets?

[Response]

To compare to the CT based method, we use the peak LAA ejection flow rate. While the overall LAA flow curve from MRI might depend on the observer, the peak LAA flow rate is clearly distinguishable and could even potentially be automated. The reproducibility of LAA flow assessment with 4D flow MRI has not been studied before, but the LAA is in a similar location as the pulmonary veins and has a comparable diameter. A study on 70 participants found good to excellent interrater reproducibility for the flow rates in the pulmonary veins (Rahman, O. et al., Radiology, 2019). We have clarified this in the introduction (page 5 line 99-102).

11) Transmitral flow evaluation is indeed an important marker used for many clinical stratifications. What is clinically more desirable using this technique, which is for image acquisition certainly faster than 4D-Flow, better than standard echo or MRI? What was the overall post-processing time for the 4D-CT?

[Response]

In this study, we wanted to investigate if it is possible to compute the transmitral and LAA flow rates from 4D CT and compare these to other MRI derived measures. So far, processing time has not been our focus. In the current setup, the manual post-processing time for the 4D-CT was 60-120 min, followed by 2-3 hours of automatic computation time per patient. We are currently working on AI based algorithms, which will reduce the manual and computational processing time substantially. We added this information to the methods.

The method proposed here measures the volume change of the left ventricle and associates positive volume change with flow through the mitral valve. It computes flow rates rather than velocities, which was found to be stronger correlated to LV remodeling than velocity-based markers (Alattar et al., Diagn. Interv. Imaging, 2022).

While the transmitral flow is already an important marker, assessment of LAA function has been challenging. When examining the LAA, this method has the advantage that it is non-invasively, in contrast to a transesophageal echocardiogram

(TEE), which is currently recommended to measure the velocities in the LAA with ultrasound. We have mentioned this in the introduction (page 3 line 54-63).

12) Any chance these patients had Doppler or tissue Doppler datasets?

[Response]

There is no doppler datasets available for these patients. However, volume flow data cannot be measured with Doppler or tissue doppler, so validation of the results from the CT motion tracking is not possible with Doppler. A recent study by Alattar et al., (Diagn. Interv. Imaging, 2022) compared the velocity in the mitral valve from 4D flow MRI with TTE and found lower transmitral velocities measured with 4D flow MRI compared to ultrasound.

In this revised manuscript, we also computed the ventricular stroke volume from short axis MRI using a volume-based approach and added this to our analysis. The comparison to the CT based stroke volume showed a smaller bias (3 ml), similar to the findings of Kaniewska et al. (Eur. Radiol. 2017), indicating that using a volume-based method in both CT and MRI leads to more similar results.

Reviewer #2

This work employs image processing techniques to extract the flow rates through the mitral orifice (MO) and across left atrial appendage (LAA) from dynamic CT (3D+time) data. A comparison of these flow rates from CT data with 4D MRI-based measurements showed reasonable agreement in a small cohort of patients (n=9) although substantial differences could be found in the MO flow during late diastole.

Although the image segmentation and registration methods employed here are well established, the study demonstrates their utility in computing flow rates from dynamic CT data which, are otherwise, not available. The manuscript is well written and methods were adequately discussed. However, the limitations of the approach are not adequately discussed.

[Response]

We have clarified the limitations of volume-based approaches in the introduction, as suggested by reviewer 1 (page 3 line 55-62):

The volume flow through the mitral valve (MV) and the left atrial appendage can, in theory, also be obtained from morphological imaging data by extracting the volume change of the main heart chambers in every frame². These volume-based approaches require high resolution 3D data with good contrast between blood pool and surrounding to achieve accurate delineation of the blood pool and myocardium, valves, papillary muscles, and myocardial trabeculae, which can be provided by 4D CT. This approach does not provide information on velocities, but quantification of flow rates. However, it was found that flow-rate derived indices correlate better with age and left ventricular remodeling compared to peak velocity based indices⁵.

Furthermore, we have clarified the description of the limitations of our approach in the limitations section of the discussion (page 16 line 256-page 17 line 276).

1. My understanding is that most patients undergoing CT also undergo ultrasound (US) evaluation. Therefore, the MO flow rate (and potentially flow across LAA?) could be easily derived from US measurements that offer a high temporal resolution. In that case, the motivation to compute these quantities from dynamic CT data should be strongly justified.

[Response]

Currently, most patient groups undergoing a CT will indeed also undergo an ultrasound investigation. Patients with, for instance, atrial fibrillation, will often even undergo a transesophageal echocardiogram (TEE), which is invasive and often requires sedation, in which the LAA velocity is an important marker. When this information can be obtained from the 4D CT exam using the method proposed, possibly even with better assessment of LAA, the TEE might, not be necessary

anymore. For patients with chest pain, which is the patient group investigated in this study, the CT is conducted to investigate the coronary arteries. Patients with a clinical suspicion of ventricular dysfunction or valvular disease often undergo an US exam but not all these patients.

The proposed approach computes flow rates rather than velocities, which was found to be stronger correlated to LV remodeling than velocity-based markers (Alattar et al. *Diagn. Interv. Imaging*, 2022). In the future, this approach could potentially replace the additional ultrasound investigation. This has been clarified in the discussion (page 17 line 277-287).

2. Both CT and MRI data employed in the current study are known to suffer from poor time resolution which could affect the computed time derivatives. Have the authors compared the computed flow rates against US data?

[Response]

Unfortunately, there is no doppler-US datasets available for these patients. However, volume flow data cannot be measured with Doppler or tissue doppler, so validation of the results from the CT motion tracking is not possible with Doppler. A recent study by Alattar et al. (*Diagn. Interv. Imaging*, 2022) compared the velocity in the mitral valve from 4D flow MRI with TTE and found lower trans mitral velocities measured with 4D flow MRI compared to ultrasound. They also compared mitral valve early and late flow rates measured with 4D flow MRI and valve tracking to 2D flow MRI and found lower flow rates with 4D flow MRI than with 2D flow MRI.

However, we added a comparison of the ventricular stroke volume measured with CT with the stroke volume measured in short axis MRI images, which is also acquired using a volume-based method. This comparison showed a smaller bias (3 ml), similar to the findings of Kaniewska et al., (*Eur. Radiol.* 2017), indicating that using a volume-based method in both CT and MRI leads to more similar results.

3. Although the study postulates that these CT-based flow measurements could be used as boundary conditions for computational fluid dynamics (CFD) models, a general CFD simulation set up for moving domain blood flow problems compute these flow rates as a by-product as opposed to employing them as boundary conditions. Applying these quantities as boundary conditions to a moving-domain problem may over constrain the problem in satisfying mass conservation.

[Response]

When simulating both the left atrium and left ventricle, accurate blood pool segmentation and wall motion is necessary, which is provided by the proposed method. When only simulating the blood flow in the left atrium, the flow rate through the mitral valve can be used as a boundary condition. We clarified this in the methods section (page 24, line 399-405) and the discussion (page 17 line 285-287).

4. *The study computes mitral flow rate from the time rate of change of the ventricular volume during diastole. However, this limits the analysis to cases when the aortic valve is fully closed and will not apply to a pathological aortic valve such as when there is aortic valve regurgitation.*

[Response]

Yes, we clarified this weakness in the limitations section (page 16 line 272-page 17 line 275):

The method proposed here measures the volume change of the left ventricle and associates positive volume change with flow through the mitral valve. Therefore, it is not applicable to patients with aortic regurgitation since it would assign backflow through the aortic valve as flow through the mitral valve.

5. *The present approach involves tedious segmentations and point-set registration techniques that can be prohibitively time-consuming. Can the authors discuss processing times for each dataset and the translational applicability of this analysis?*

[Response]

As mentioned to reviewer 1, in the current setup, the manual post-processing time for the 4D-CT was 60-120 min, followed by 2-3 hours of automatic computation time per patient. In the future, using AI based algorithms, we believe that the manual and computational processing time can be reduced substantially, which would be necessary for clinical use. We added this information in the methods, page 22 line 360-361).

6. *Page 10, Discussion, while explaining the differences in stroke volume between CT and MRI compared to an earlier meta analysis, it is noted that the present MRI-based stroke volume is based on integrating flow rate through the mitral orifice and could potentially explain the large difference from CT data. This only raises concerns on the accuracy of the MRI-based velocities and flow rates as both methods should theoretically converge to the same value. On the other hand, can the authors use ventricular volumes from MRI data and compare the stroke volume against CT? This should be straightforward as the volumes could be easily extracted from 4D MRI.*

This poses two issues - ideally both methods should give exact results (volume-based or integral of flow rate). If the MRI-based volumes

[Response]

Thank you for this suggestion. Beside 4D flow MRI, short axis cine images were acquired, which we used to compute the ventricular stroke volume for this revision. We found a lower bias when comparing the stroke volume from the short axis MRI to CT of 3 ml, which is similar to what was found in the meta-analysis you are referring to. There is a relatively large difference between the two MRI based methods (bias 11 ml), hinting towards structural differences in flow-based vs volume-based methods, which is similar to what has been reported by Kamphuis et al. (JMRI, 2018).

7. *The motivation to perform a separate registration of LAA is not discussed.*

[Response]

We added the need to perform a separate registration of the LAA, which increased the manual input time, to the limitations (page 16 line 262-264):

To track the relatively fast motion of the LAA, it was tracked separately, since then it can be deformed independently of the LA. This increased the processing time but was necessary to properly track the motion.

Minor:

- 1. Caption of Fig. 2 should be "Flow through..."*
- 2. Figure captions need to be properly formatted for subfigures (A, B, etc.).*
- 3. Several citations in the manuscript are not properly referenced. Please fix them.*
- 4. Typo in Line 156, Page 10: "intervall"*

[Response]

We have made the suggested minor changes.

Reviewer #3

You are assuming that your deformable registration is perfect here which is never the case. (Referring to “Using an algorithm to track the motion based on only one manual segmentation allows for more automatic and accurate processing of the data and would also facilitate calculations of other parameters relative to the intracardiac blood flow dynamics”)

[Response]

We removed “and accurate” from the sentence.

You are not deriving flow fields from 4D-CT. What you are calculating the change in volume of LV, LA, and LAA. You are then relating it to volume flow rate assuming that there is no leakage from other orifices (ex: zero aortic valve back flow in LV) connected to the volume you are tracking, which is suspect. Furthermore, 4D-Flow MRI is not going to give you accurate results because of volume averaging. So you are comparing your method to a method that you know is not accurate.

[Response]

We clarified in that sections that flow rates and not flow fields are computed (page 5 line 105-107):

In this paper we calculate the transmitral and LAA flow rate from 4D CT using a motion tracking algorithm. We compared the flow rates derived from 4D-CT with 4D flow MRI derived LAA and mitral flow rates and cine MRI derived LV stroke volume.

Additionally, we changed the title of the manuscript from ‘Assessment of **mitral valve** and left atrial appendage flow rate from cardiac 4D-CT’ to ‘Assessment of **transmitral** and left atrial appendage flow rate from cardiac 4D-CT’.

Furthermore, we also clarified in the limitations that this approach is assuming no valve regurgitations. Since there is no clear ground truth, we changed all statements from “validation with 4D flow MRI” to “comparison to 4D flow MRI” and clarified in the limitations that there is no gold standard to compare to.

To have another reference point, we computed stroke volumes from short axis MRI and compared them to the results from CT and 4D flow MRI (Figure 3). When comparing the stroke volume from CT with the volume-based MRI approach, the bias was only 3 ml, which is similar to what was found by Kaniewska et al., (Eur. Radiol. 2017).

As I understand, 4D-Flow MRI essentially tracks MOVING spins by encoding the velocity in the phase. If you have non-fluid tissue moving, as is the case with the mitral valve, how is it affecting the accuracy of you mitral valve volumetric flow calculation?

[Response]

4D flow MRI tracks indeed spins/protons for about 1 ms and calculates from all spins the velocity in a voxel. Valve tissue and myocardium also contain protons. The

velocity in the voxels covering the valve tissue might theoretically be slightly affected by this, but the effect on the flow through the whole mitral valve, which is integrated from many voxels, is negligible. The motion of the mitral valve is tracked based on cine bSSFP (balance steady-state free-precession) image (3chamber view). We have clarified this process in the methods section (page 25, line 416-421):

The valve annulus movement was automatically tracked on a cine bSSFP (balance steady-state free-precession) image (3chamber view) over the cardiac cycle, and from the resulting positions reformatted planes were placed in the 4D flow volume. Subsequently, the volumetric stroke volume was calculated by integrating the velocity over the mitral valve orifice, and correcting for through-plane motion, similarly to 19.

I can image when the mitral valve is fully open, you probably get about 30 voxels of 4D-Flow MRI across the diameter at best. At the beginning and end of diastole, how big is the opening in the number of 4D-Flow MRI voxels. Can you rely on the calculation?

[Response]

The mitral valve has a diameter of ca 30 mm. The spatial resolution of the MRI measurement was 2.9 mm in each direction, following the recommendation for 4D flow MRI (Dyverfeldt JCMR 2015), leading to ca. 10 voxels along the diameter. Studies have demonstrated that 5–6 voxels across a vessel diameter are sufficient for flow volume quantification (Hofman, MRM 1995). Alattar et al. (Diagn. Interv. Imaging, 2022) have performed a comparison of the MV flow rate and velocity measured in 4D flow MRI with 2D flow MRI and TEE and found that 4D flow is suitable to measure flow rates in the mitral valve.

How do you explain this? (Referring to “. For all patients except patient 6, the early diastolic peak flow was higher in the 4D-CT than in 4D flow MRI.”)

[Response]

We extended the discussion of the potential reasons for the differences between CT and 4D flow MRI (page 13 line 208-page 14 line 224):

The flow rates during late diastole agree better than during early diastole, both in the LAA and in the mitral valve. For patient 3, 4 and 8, the CT based method shows a peak in the LAA contraction during early diastole, which is not visible in 4D flow MRI. For these patients, the E peak of the MV flow curve is also higher in the CT based method than in 4D flow MRI. Alattar et al. also found lower MV flow rates in 4D flow MRI compared to 2D flow MRI, however, they found similar agreement between early and late diastolic flow 5. One possible explanation for the difference between early and late diastole could be that 4D MRI is computed as the average of many heart beats, that are composed to one image based on the R wave in the ECG signal, which corresponds to end diastole. Each measured cardiac cycle is normalized to the average heart rate to compose the image. Changes in the heart rate alter the length of diastasis, which then alters the relative position

*of the E wave in the heart rate-normalized signal, which could lead to a stronger smoothing of the signal during this time.
Another possible explanation is that the 4D flow MRI was acquired during free breathing, while the CT was acquired during an inspiration breath hold and the short axis MRI was acquired during end-expiration breath hold. This could alter the pressure in the lungs, which influences the pressure in the left atrium and might alter the flow through the mitral valve.*

And this could very well be because in your method you are not accounting for back flow from say aortic valve and the inherent error due to volume averaging (Referring to ‘ This can lead to slightly lower stroke volumes compared to a volume based approach’)

[Response]

In this study, we excluded patients with valvular diseases. We have double checked the flow rates at the aortic valve in 4D flow MRI and there is no back flow. We included to the limitations that this method cannot be applied to patients with aortic regurgitation. The error of volume averaging in CT is comparably small, since the voxel size is only 0.35 mm. We added the discussion of the impact of the trabeculae in the LV on the volume-based computation in CT to the manuscript. The flow in 4D flow MRI is computed as the integral of the volume flow in the MV plane, which is not that sensitive to volume averaging, since the flow at the walls is very small compared to the flow in the center of the valve. We have clarified the process of the velocity measurement at the mitral valve, as cited in the previous answer.

The main limitation is that you are comparing results from segmented 4D CT with 4D Flow MRI, both of which are flawed due to the reasons indicated previously. You do not really know what is actually occurring in reality, i.e., you have no idea what the ground truth is.

[Response]

In the revised manuscript, we added a third approach, where we computed the stroke volume from short axis MRI, also with a volume-based method. As mentioned previously, the bias between the volume-based methods (CT and MRI) was only 3 ml, which is similar to what was found by Kaniewska et al., (Eur. Radiol. 2017). Furthermore, we changed ‘validation using 4D flow MRI’ to ‘comparison with 4D flow MRI’ and we added the lack of a ground truth in the limitations section in the discussion.

Feasible, yes. Useful? I don't know. [Referring to 'In conclusion, calculating the flow through the mitral valve and the LAA orifice from 4D-CT is feasible. ']

[Response]

To point out the potential usefulness of computing flow rates in the LAA and the mitral valve, we added this paragraph to the discussion (page 16 line 272-page 17 line 287):

The method proposed here measures the volume change of the left ventricle and associates positive volume change with flow through the mitral valve. Therefore, it is not applicable to patients with aortic regurgitation since it would assign backflow through the aortic valve as flow through the mitral valve. Despite that, the method computes flow rates rather than velocities, which was found to be stronger correlated to LV remodeling than velocity-based markers 5.

When examining the LAA, this method has the advantage that it is non-invasive, in contrast to a transesophageal echocardiogram (TEE), which is currently recommended for assessment of the LAA. Extracting transmitral and LAA flow rate from CT could, in the future, reduce the need for using additional imaging exams in some patient groups, as, for instance, the TEE exam in patients with atrial fibrillation before ablation treatment. For patients with an indication for a CT angiography, as studied in this project, the addition of the presented approach would enable accurate assessment of both diastolic and systolic ventricular function, besides the assessment of the coronary arteries, potentially removing the need to perform some ultrasound investigations. The derived wall motion can also be used as boundary conditions in CFD simulations of the heart, while the transmitral or LAA flow rates could facilitate simulation of the blood flow in only the left atrium or LAA.

You need to validate the method first using a in vitro flexible phantom with realistic geometry where you can control the actual flow rates and check for accuracy before you conduct this study.

[Response]

We suppose that the difference in the measurements from CT and MRI originates from several reasons, which are difficult to include in an in-vitro phantom. 4D flow MRI is averaging the flow over many cardiac cycles with differing length and other physiological variations, leading to a stronger smoothing in this data set. The segmentations of the left ventricle do not include trabeculations or papillary muscles. However, it is possible that the trabeculated structures appear brighter during end diastole, when the ventricle is at its largest, compared to the end systolic phase, when the ventricle is most compressed. This could increase the estimated stroke volume from CT.

The 4D flow MRI was acquired during free breathing while the CT was acquired during a breath hold. These differing breathing patterns could lead to different pulmonary pressure and thus influence the blood flow in the left atrium.

We do not think that an in vitro flexible phantom would be able to represent these differences in a meaningful way and add useful information to this manuscript. As mentioned before, due to the lack of a ground truth, we changed the manuscript from a validation to a comparison of two methods. Furthermore, we also added a third method for computing the ventricular stroke volume from short axis MRI and found good agreement to the CT based stroke volume.

How can you assume that some source points are locked? The entire boundary is moving.

[Response]

We assume that the edge of the pulmonary veins and the aorta, where the segmentation ends, do not move. This is a simplification; however, it does not affect the motion of the left atrial appendage or the left ventricle, thus it does not have any effect on the presented results. We clarified this in the methods (page 22 line 362-365).

How much error did smoothing introduce? How did it impact your subsequent registration and volume calculations?

[Response]

The surface in the left atrial appendage and the left ventricle is trabeculated with structures smaller than the spatial resolution of the CT acquisition. The registration is keeping the topology of the source surface, thus topological changes cannot be considered by the registration. The trabeculations lead to the larger average distance between the registration and target surface, as shown in Figure 1. It is possible the volume calculation overestimates the stroke volume slightly, since the trabeculated structures in the LV might be considered as part of the blood volume at end diastole, when they are surrounded by blood, while being excluded from the volume at end systole, when the ventricle is fully contracted. We added this consideration to the discussion (page 14 line 225-232).

How did you come up with 250 Hounsfield units? Furthermore, this seems to be a lot of MANUAL work for each time step. Not only is this time consuming, it is very susceptible to operator variability. The only thing that is automatic here is the registration.

[Response]

As presented in Figure 1, enhanced blood pools have a hounsfield unit of 300-500, while contrast enhanced myocardium has a hounsfield unit of 80-140, thus a threshold of 250 is sufficient to separate the blood pool from its surrounding structures. We added a reference to the book chapter 'Computed Tomography' in Cardiac CT Imaging: Diagnosis of Cardiovascular Disease by Matthew J. Budoff to the manuscript (page 21 line 348).

Table 1.1 Typical Hounsfield unit values

Air ~ -1000 HU
Fat -100 to -40
Water - zero
Non-enhanced myocardium and blood - 40-60
Contrast enhanced myocardium 80-140
Calcium >130 (to about 1000)
Enhanced blood pools (lumen, aorta, LV) 300-500
Metal >1000

Figure 1: Table with typical Hounsfield units from Budoff, Matthew J. 'Computed Tomography'. In *Cardiac CT Imaging Diagnosis of Cardiovascular Disease*, 3rd ed., page 4. Springer Cham. <https://doi.org/10.1007/978-3-319-28219-0>.

The in-house code developed in MevisLab, that was used to create the target surfaces, requires only the file path of the CT images and 10 seed points at one time step to create threshold-based segmentations of all time steps. After this, planes cutting the surfaces at the pulmonary veins and the aorta were placed manually. Since these planes were static, this only needed to be done in the first time-step. We added the overall manual processing time of 60-120 minutes per patient to the manuscript (page 21, line 352-353). We expect that this time can be considerably reduced using further automation.

This assumption is not true. These points are in fact moving. (Referring to "The 267 boundary points at the pulmonary veins and the aorta were set to be non-moving, which eases 268 the definition of boundary conditions in later CFD simulations")

[Response]

As mentioned above, this assumption does not affect the motion of the LAA or the LV and this does not affect the results.

How did you come up with this scheme? Why did you start at 100, why not say 110? Why did you lower by 1 in 100 step. Why not 0.5. This scheme seems pretty arbitrary without any justification. May not be repeatable for other data sets. Furthermore, it probably impacts the accuracy of your results.

[Response]

In the article from Amberg et al, describing the algorithm it is written: "For all methods a gradual relaxation of the stiffness constraint lowering α in 100 equally distributed steps was used. The absolute α values depend on the template shape and resolution and should be chosen such that at the beginning of the algorithm only global deformations are recovered. The lowest possible alpha depends also on the

type of data. If α becomes too low the conditioning of A suffers and the solution becomes unstable. All experiments were done with the same minimal alpha, staying far away from an unstable system. We start with an excessively high alpha, because starting with a high alpha cannot decrease the quality of the results, it may only lead to more steps being necessary.”

Internal testing during the implementation showed that a start value of 100 is sufficiently large to only cover global transformations and that the end value of 1 for the registration of the full left heart was sufficient to resemble more detailed deformations while the solution was stable.

Due to the smaller geometry of the left atrial appendage, alpha was only lowered to 10 to ensure stability. To speed up the registration, for the LAA alpha was lowered in only 20 steps. We clarified this in the methods section (page 23 line 382-383).

How did you come up with 10?

[Response]

The landmark term should ensure that the landmark points do not move. A value of 10 increases the weight of the landmark terms enough to outweigh any other forces acting on these points. We clarified this in the methods section (page 23-24 line 390-391).

How did you come up with 15 knots. Why not 14 or 16?

[Response]

The number of knots defines the degrees of freedom of the spline fit and regulates the smoothing. The input data consists of 20 datapoints, to smooth the noisy input data, we found that 15 knots lead to a decent smoothing while preserving the overall signal. There is no large difference when taking 14, 15 or 16 knots; however, 20 knots lead to oscillations in the flow curve while 10 knots dampen the signal too much. We clarified this in the methods section (page 24 line 401-401).

What numerical scheme did you use? How did you handle noise in the data? What was the truncation error?

[Response]

The volume of the closed triangulated surface representing the LV and LAA respectively, was calculated using the divergence theorem. Since the motion was interpolated and evaluated with a temporal resolution of 5 ms, there were 150-250 data points, depending on the heart rate of the patient. To differentiate the volume by time, a spline was fitted through the volume points using the SPLINEFIT tool and then differentiated using the pddiff function of the SPLINEFIT tool. Due to the large number of data points, the spline fit tool computed similar flow rates as a central differences scheme. We clarified this in the methods section (page 24 line 407).

What this means is that the volumetric change you are calculating is not only because of flow of blood into LAA but also wholesale motion of LAA since your measurement planes are static.

[Response]

We used a static plane to compute the flow rates from CT in order to facilitate the comparison with 4D flow MRI, and thus both measurements include the LAA contraction and overall motion. Due to that the algorithm computes the wall motion in CT, it is also possible to use a moving plane to compute flow rates in the LAA. In future studies that do not compare to 4D flow MRI, but rather relate the LAA flow rates to other clinical parameters, this is probably the more desired approach. We have clarified this in the discussion (page 13 line 203-208).

Decision letter and referee reports: second round

6th Oct 22

Dear Professor Ebbers,

Your manuscript entitled "Assessment of Transmitral and Left Atrial Appendage Flow Rate from Cardiac 4D-CT" has now been seen again by our referees, whose comments appear below. In light of their advice I am delighted to say that we are happy, in principle, to publish a suitably revised version in Communications Medicine under the open access CC BY license (Creative Commons Attribution v4.0 International License).

We therefore invite you to revise your paper one last time to address the remaining concerns of Reviewer 2. At the same time we ask that you edit your manuscript to comply with our policy and format requirements and to maximise the accessibility and therefore the impact of your work.

Please note that it may still be possible for your paper to be published before the end of 2022, but in order to do this we will need you to address these points as quickly as possible so that we can move forward with your paper.

* Please see the attached document for editorial requests for the final version (.docx file). Please ensure a completed version of this file is uploaded as a Related Manuscript with your final submission.

* Please review our [final submission file checklist](https://www.nature.com/documents/commsj-file-checklist.pdf) to ensure all necessary files are present with your final submission and to avoid delays in accepting your manuscript. For your reference, a style and formatting guide is available [here](https://www.nature.com/documents/commsj-life-style-formatting-guide-accept.pdf) and includes all of our style requirements. We also have a [style and formatting checklist](https://www.nature.com/documents/commsj-life-style-formatting-checklist-article.pdf), should you find this a useful resource.

It is important that you pay careful attention to the requests in these documents to avoid a delay in formal acceptance of the article.

Open access

Communications Medicine is a fully open access journal. Articles are made freely accessible on publication under a [CC BY license](http://creativecommons.org/licenses/by/4.0) (Creative Commons Attribution 4.0 International License). This license allows maximum dissemination and re-use of open access materials and is preferred by many research funding bodies. For further information about article processing charges, open access funding, and advice and support from Nature Portfolio, please visit our [website](https://www.nature.com/commsmed/about/article-processing-charges).

At acceptance, you will be provided with instructions for completing this CC BY license on behalf of all authors. This grants us the necessary permissions to publish your paper. Additionally, you will be asked to declare that all required third party permissions have been obtained, and to provide billing information in order to pay the article-processing charge (APC).

Please use the following link to upload your revised files:
[link redacted]

We hope to hear from you within two weeks; please let us know if the process may take longer.

Congratulations on an excellent paper!

Best regards,

Andreia Cunha, PhD
Chief Editor
Communications Medicine

on behalf of

Dr Betty Raman
Editorial Board Member
Communications Medicine

PS: At acceptance, the corresponding author will be required to complete an Open Access Licence to Publish on behalf of all authors, declare that all required third party permissions have been obtained and provide billing information in order to pay the article-processing charge (APC) via credit card or invoice. Please note that your paper cannot be sent for typesetting to our production team until we have received these pieces of information; **therefore, please ensure that you have this information ready when submitting the final version of your manuscript.**

REVIEWERS' COMMENTS:

Reviewer #1 (Remarks to the Author):

Thank you for the revised version and nicely addressed comments. Thank you so much for addressing minor points ranging from the decimal point correction to additional data background sounds stupid info in the introduction. Thank you as well for addressing my questions regarding the diastolic measurements and overall clinical translation.

Reviewer #2 (Remarks to the Author):

The authors have adequately addressed my concerns in the review. There are only some minor concerns that I have described below and can be taken care of during the final revision. I recommend publishing the manuscript.

1. CFD analysis in the LA has been performed using image-based models by Juan C. del Alamo's group, Vedula et al., and a few others. See for e.g.,

<https://doi.org/10.3389/fphys.2021.596596>
<https://pubmed.ncbi.nlm.nih.gov/26329022/>

Likewise, these groups and a few others have also employed nonlinear image registration techniques to automate displacement extraction. These papers should be cited at appropriate locations in the manuscript.

2. Grammar and sentence construction in the newly added text need to be rectified. For e.g.,

- Line 261 should be 'automated' and not 'automized'.
- Line 271, 'which was found to be stronger correlated to LV remodeling than velocity-based markers' should be 'strongly correlated' and 'than velocity-based markers.'

Reviewer #3 (Remarks to the Author):

All comments have been address adequately.

COMMSMED-21-0591-T

Assessment of mitral valve and left atrial appendage
flow rate from cardiac 4D-CT

Response to Referees – Final Revision

REVIEWERS' COMMENTS:

Reviewer #1 (Remarks to the Author):

Thank you for the revised version and nicely addressed comments. Thank you so much for addressing minor points ranging from the decimal point correction to additional data background sounds stupid info in the introduction. Thank you as well for addressing my questions regarding the diastolic measurements and overall clinical translation.

Reviewer #2 (Remarks to the Author):

The authors have adequately addressed my concerns in the review. There are only some minor concerns that I have described below and can be taken care of during the final revision. I recommend publishing the manuscript.

1. CFD analysis in the LA has been performed using image-based models by Juan C. del Alamo's group, Vedula et al., and a few others. See for e.g.,

*<https://doi.org/10.3389/fphys.2021.596596>
<https://pubmed.ncbi.nlm.nih.gov/26329022/>*

Likewise, these groups and a few others have also employed nonlinear image registration techniques to automate displacement extraction. These papers should be cited at appropriate locations in the manuscript.

Thank you for this advice. We added the suggested references in the introduction, Page 5 line 9 to 16 and changed in the discussion, page 26 line 13 to 16.

2. Grammar and sentence construction in the newly added text need to be rectified. For e.g,

- Line 261 should be 'automated' and not 'automized'.

- Line 271, 'which was found to be stronger correlated to LV remodeling that velocity-based markers' should be 'strongly correlated' and 'than velocity-based markers.'

Thank you for noticing this, we adjusted the text.

Reviewer #3:

All comments have been address adequately.